# Risk factors for aggravated COVID-19 despite medical care after admission among Japanese patients: A Japanese association for infectious diseases COVID registry study

Itaru Nakamura[1¶], Yusuke Koizumi[2¶], Hideki Araoka[3¶*], Hiroaki Hata[4¶], Takashi Miki[5], Yusuke Tanaka[5], Chie Kobayashi[5], Kazuyoshi Kawakami[6¶]

**1** Department of Infection Prevention and Control, Tokyo Medical University Hospital, Tokyo, Japan, **2** Department of Infection Prevention and Control, Wakayama Medical University, Wakayama, Japan, **3** Department of Infectious Diseases, Toranomon Hospital, Tokyo, Japan, **4** Department of Surgery, National Hospital Organization Kyoto Medical Center, Kyoto, Japan, **5** Astellas Pharma Inc., Tokyo, Japan, **6** Hirose Hospital, Miyagi, Japan

¶ Ad Hoc Committee for the COVID-19–Aggravating Factors Search Project of the Japanese Association for Infectious Diseases.
* h-araoka@toranomon.gr.jp

## Abstract

### Background

Limited data are available on COVID-19 outcomes in the Japanese population. During the Wuhan strain–dominant period, we investigated factors associated with clinical deterioration of COVID-19 despite inpatient medical care in Japan.

### Materials and methods

This retrospective, multicenter cohort study used data from the Japanese Association for Infectious Diseases (JAID) COVID registry to identify risk factors for progression to severe disease after hospital admission. The study population included Japanese patients with a confirmed diagnosis of COVID-19 between January 2020 and March 2021. Baseline data included demographics, symptoms at disease onset and admission, and laboratory findings. Univariate and multivariate logistic regression models were used to calculate odds ratios (ORs) and 95% confidence intervals (CIs) for each explanatory variable associated with disease aggravation. Multiple imputation was used to handle missing data. The predictive performance of two multivariate models—Model I (based on measurable clinical findings) and Model II (based on interview data)—was assessed using the area under the receiver operating characteristic curve (AUC).

### Results

A total of 2,884 patients were enrolled across 36 institutions in Japan. After excluding 25 ineligible cases and 45 protocol deviations, 2,814 patients were included in the

**Data availability statement:** All relevant data are within the manuscript and its Supporting Information files.

**Funding:** This JAID-COVID registry study was conducted as a JAID joint research project with research funding from Astellas Pharma.

**Competing interests:** The authors have declared that no competing interests exist.

analysis. Multivariate analysis identified several independent risk factors for disease aggravation despite inpatient care, including admission early in the pandemic, older age, impaired consciousness, respiratory distress, elevated BMI, high body temperature, anemia, cholestasis, dehydration, and elevated γ-GTP and LDH levels. The AUCs were 0.8928 for Model I and 0.8862 for Model II, indicating strong discriminatory power.

## Conclusion

This study identified key risk factors for COVID-19 progression despite inpatient medical care among Japanese patients. These findings underscore the importance of early risk stratification for patients at high risk of deterioration despite treatment and may inform preparedness strategies for future respiratory pandemics.

## Introduction

Coronavirus disease 2019 (COVID-19) is caused by severe acute respiratory syndrome coronavirus 2 (SARS-CoV-2). The global mortality rate was particularly high at the onset of the pandemic, dominated by the Wuhan strain and subsequently by the Alpha and Delta variants. Mortality rates in the Japanese population have been reported to be among the lowest worldwide. However, during the Wuhan strain period—when vaccines were not yet available—some patient groups in Japan experienced disease worsening associated with common factors and comorbidities.

Previous studies from multiple countries have identified various factors influencing COVID-19 prognosis. Reported risk factors include age, sex, number and type of comorbidities, respiratory rate, body mass index (BMI), peripheral oxygen saturation, and level of consciousness, as measured by tools such as the Glasgow Coma Scale (GCS). Laboratory parameters reported as prognostic indicators include white blood cell (WBC) count, lymphocyte count, blood glucose, albumin (ALB), aspartate aminotransferase (AST), lactate dehydrogenase (LDH), creatinine kinase, urea, C-reactive protein (CRP), troponin I, procalcitonin, and D-dimer [1–10]. However, these risk factors have generally been associated with patients already in severe condition upon hospital admission rather than with disease aggravation occurring despite medical care after admission. Moreover, most studies investigating risk factors for severe disease progression or death in COVID-19 patients have focused on non-Japanese populations. Data on COVID-19 outcomes specific to Japanese patients remain limited to a few reports [11–14].

In this study, we aimed to identify factors associated with COVID-19 aggravation during inpatient treatment despite receiving medical care, focusing on the Wuhan strain–dominant period in an unvaccinated Japanese population.

## Materials and methods

### Patients and study design

This retrospective, multicenter cohort study was conducted to investigate factors associated with disease aggravation in Japanese patients with COVID-19. The study

utilized the COVID registry of the Japanese Association for Infectious Diseases (JAID), which includes data from 36 institutions across Japan. Study participants were Japanese patients diagnosed with COVID-19 over a 15-month period from January 2020 to March 2021. Three waves of COVID-19 occurred during this timeframe in Japan: the first wave of the Wuhan strain from January 1 to May 27, 2020; the second wave of the Wuhan strain from May 28 to September 25, 2020; and the third wave of the Wuhan strain from September 26, 2020, to March 31, 2021. These waves were defined in this study as Periods I, II, and III, respectively.

The primary objective was to identify the presence or absence of COVID-19 aggravation after hospital admission despite medical care. Secondary objectives included characterizing the clinical course and outcomes of COVID-19 in the Japanese population.

### Inclusion and exclusion criteria

Inclusion criteria comprised Japanese patients with a definitive COVID-19 diagnosis confirmed by nucleic acid amplification test (polymerase chain reaction or loop-mediated isothermal amplification) or immunochromatography, who completed inpatient treatment and were either discharged, transferred, or died during the 15-month study period. Patients who declined to provide medical information for research purposes were excluded.

### Patient data

Patient demographics and treatment data were collected using electronic case report forms managed by facility research coordinators at 36 hospitals nationwide. For some hospitals, data from the COVID-19 REGISTRY JAPAN (COVIREGI-JP) maintained by the National Center for Global Health and Medicine were incorporated with permission. Baseline data included demographics, initial COVID-19 symptoms at admission, and laboratory findings. Baseline was defined as (i) data collected on the admission date (or, if unavailable, shortly thereafter but prior to severe disease development) for patients newly hospitalized with COVID-19, or (ii) the latest data before COVID-19 onset for patients who acquired the infection nosocomially.

Collected independent variables included age, sex, body mass index (BMI), pregnancy status, physical findings on admission (body temperature, circulatory and respiratory status, level of consciousness measured by GCS or Japan Coma Scale [JCS]), laboratory findings, smoking status, and comorbidities such as cardiovascular, respiratory, renal, allergic, or immunodeficiency diseases (including HIV or primary immunodeficiency), collagen disease, malignant tumors, chemotherapy, radiotherapy, and medications for underlying conditions.

Outcomes examined included aggravation (defined as requiring tracheal intubation, mechanical ventilation, extracorporeal membrane oxygenation [ECMO], initiation of blood purification therapy, intensive care unit [ICU] admission, or death), timing of hospital admission, COVID-19-associated complications, and disease duration (from symptom onset to recovery). Case report forms were registered between April 1 and December 31, 2021. All data were de-identified and anonymized.

### Sample size calculation

The primary aim of this study was exploratory, to assess clinical characteristics and risk factors associated with COVID-19 severity in Japan; thus, no a priori hypothesis testing was performed. Sample size was determined based on feasibility, estimating approximately 2,000 patients could be registered.

According to published data from the national epidemiological surveillance of infectious diseases system and active epidemiological surveillance during the study period, rates of ICU admission, invasive ventilation (including endotracheal intubation), death, and ECMO use were 35/323 (11%), 49/347 (14%), 10/516 (2%) deaths, and 18 cases, respectively [15]. Consequently, the severity rate was conservatively estimated at 10–15%. Assuming 2,000 patients were enrolled, approximately 200–300 cases of severe COVID-19 aggravation were expected.

## Statistical analysis

The full analysis set (FAS) included all registered patients after applying exclusion criteria. For statistical analysis, a modified full analysis set (mFAS) was created by excluding patients who exhibited disease aggravation on the day of admission or before admission. Background characteristics and risk factors for worsening after admission were analyzed using the mFAS, as the focus was on predicting outcomes among patients who were not initially severe but who worsened despite medical care.

Univariate logistic regression models were applied to the mFAS to estimate odds ratios (ORs) and 95% confidence intervals (CIs) for each explanatory variable regarding the presence or absence of disease aggravation. Unadjusted ORs and 95% CIs were reported. Continuous variables were standardized as deviation scores. Spearman's correlation analysis was performed to evaluate associations between platelet-to-lymphocyte ratio (PLR) [16,17], neutrophil-to-lymphocyte ratio (NLR) [16,18], and laboratory findings, with correlograms constructed for all clinical test variables.

Two multivariate logistic regression models were developed using mFAS to assess the association of variables with aggravation risk. Model I estimated risk based on objective measures such as laboratory data and other measurable variables available at admission without requiring patient interviews. Model II estimated risk based on patient interviews at admission and information from referring physicians when laboratory data were unavailable. Variables with p-values <0.1 and missing data rates <30% from univariate analyses, alongside clinically relevant variables, were included in both models. The area under the receiver operating characteristic (ROC) curve (AUC) was calculated to evaluate the discriminatory power of each model.

Although the forward stepwise method is commonly used for variable selection, the presence of variable missingness and varying missing rates raised concerns about model instability and bias from automated selection. Therefore, variables considered clinically important or with significant p-values were selected a priori, and a full-model approach including all these variables was adopted. Adjusted ORs and 95% CIs were reported. Multiple imputation was employed to handle missing data. Statistical significance was defined as p < 0.05.

## Ethical considerations

This study was approved by the Tohoku University Ethics Committee (approval number T2020-0161) and the institutional review boards of all collaborating institutes (UMIN registration number 000043802). The requirement for participant consent was formally waived due to the retrospective and exploratory nature of the cohort study. Patient data were anonymized and extracted from medical records. An opt-out notice was posted on each institution's website to inform patients of the study and provide an opportunity to withdraw.

## Results

A total of 2,884 patients were enrolled from 36 institutions across Japan participating in the JAID-COVID registry. After excluding 25 ineligible patients and 45 with protocol deviations, 2,814 cases were included in the full analysis set (FAS). To identify factors associated with clinical aggravation despite medical care, patients who were already severe before admission (n = 45) and those who were severe at the time of admission (n = 211) were excluded. As a result, 2,598 patients were included in the modified full analysis set (mFAS), which served as the primary analytical cohort for this study (Fig 1).

The demographic characteristics of the mFAS population are presented in Table 1. Among the 2,598 patients, 1,520 were male and 1,078 were female. The mean age was 57.2 (standard deviation [SD] = 21.2), and the median age was 58 years. Regarding the timing of hospital admission, 483 patients were admitted during Period I, 716 during Period II, and 1,399 during Period III. Overall, 170 patients (6.5%) experienced disease aggravation: 44 (9.1%) in Period I, 25 (3.5%) in Period II, and 101 (7.2%) in Period III.

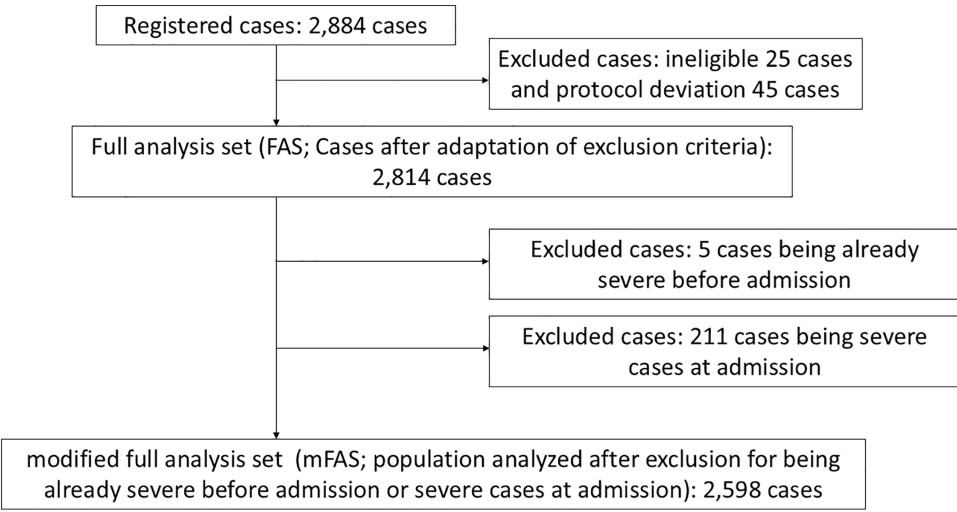

**Fig 1. Flow diagram of the study cohort.** Registered cases: n = 2,884 Excluded cases: ineligible (n = 25), protocol deviation (n = 45). Full analysis set (FAS): n = 2,814.

The crude mortality rate was 3.8% (98/2,598), comprising 28 deaths in Period I (5.8%), 14 in Period II (2.0%), and 56 in Period III (4.0%). The average time from symptom onset to hospital admission was longest in Period I (8.1 days), compared to Period II (6.1 days) and Period III (6.6 days).

Treatments administered during hospitalization included corticosteroids in 1,058 patients (40.7%), comprising methylprednisolone (n = 136), prednisolone (n = 65), hydrocortisone (n = 8), dexamethasone (n = 730), betamethasone (n = 2), and ciclesonide (n = 206). Antiviral drugs were given to 1,066 patients (41.0%), including remdesivir (n = 537) and favipiravir (n = 594). Additional interventions included oxygen therapy (n = 957; 36.8%), mechanical ventilation (n = 111; 4.3%), ECMO (n = 5; 0.19%), blood purification therapy (n = 8; 0.31%), catecholamines (n = 50; 1.9%), and anticoagulant therapy (n = 545; 21.0%).

Modified full analysis set (mFAS): n = 2,598 (excluding cases already severe before or at admission) Univariate analysis of patient demographics in the mFAS revealed statistically significant associations between clinical aggravation and the following variables: age, sex, smoking history, timing of admission, presence of underlying diseases, and administration of medications for underlying conditions (S1 Table). On the other hand, no significant associations were found for pregnancy, immunodeficiency syndromes (including HIV), allergic diseases, asthma, a history of cured malignancy, or the use of calcineurin inhibitors, anticoagulants, estrogen agonists, immune checkpoint inhibitors, or anti-allergy medications.

Univariate analysis of clinical symptoms demonstrated that fever, respiratory distress, and elevated body temperature were significantly associated with increased odds of disease aggravation (S2 Table). Lower levels of consciousness on admission, as reflected by JCS Categories I, II, and III and reduced GCS scores, were also significant predictors. In contrast, the presence of olfactory or taste disturbances was associated with significantly lower odds of aggravation.

Regarding the laboratory findings at admission (S3 Table), several variables were significantly associated with disease aggravation in the univariate analysis. These included WBC, hemogram, red blood cell count (RBC), hemoglobin (Hb), hematocrit (Ht), total protein (TP), albumin (ALB), gamma-glutamyl transpeptidase (γGTP), lactate dehydrogenase (LDH), blood urea nitrogen (BUN), creatinine (Cr), C-reactive protein (CRP), hemoglobin A1c (HbA1c), activated partial thromboplastin time (APTT), fibrinogen, and ferritin. Procalcitonin levels, however, were not statistically significant in predicting

**Table 1. Patient Demographics (mFAS).**

| Variable | | N | % |
|---|---|---|---|
| Total patients | | 2,598 | |
| Severe disease after admission | | 170 | 6.5 |
| Death after admission | | 98 | 3.8 |
| Age (years) | N | 2,598 | |
| | Mean | 57.2 | |
| | Median | 58 | |
| | Standard deviation | 21.2 | |
| Sex | Male | 1,520 | 58.5 |
| | Female | 1,078 | 41.5 |
| BMI | <18.5 | 247 | 10.5 |
| | 18.5–24.9 | 1,338 | 56.8 |
| | 25–29.9 | 586 | 24.9 |
| | 30–34.9 | 132 | 5.6 |
| | 35–39.9 | 33 | 1.4 |
| | 40–44.9 | 15 | 0.6 |
| | ≥45 | 4 | 0.2 |
| | Missing value | 243 | |
| Smoking | Yes/No | 1,184/906 | 45.6/34.9 |
| | Missing value | 508 | 19.6 |
| Pregnancy | Yes/No | 27/1,042 | |
| | Missing value | 9 | |
| Reason for admission | First admission for COVID-19 | 2,184 | 84.1 |
| | Transfer due to COVID-19 | 317 | 12.2 |
| | Nosocomial infection | 77 | 3.0 |
| | Others and missing value | 18 | |
| Symptoms at disease onset | Fever | 1,832 | 70.5 |
| | Cough | 915 | 35.2 |
| | Fatigue | 673 | 25.9 |
| | Olfactory disturbance | 138 | 5.3 |
| | Taste disturbance | 175 | 6.7 |
| | Breathing difficulty | 258 | 9.9 |
| | Consciousness disturbance | 20 | 0.8 |
| | Diarrhea | 136 | 5.2 |

BMI; Body mass index (kg/m²).

aggravation in this cohort. The correlogram showing the correlations between the platelet-to-lymphocyte ratio (PLR), neutrophil-to-lymphocyte ratio (NLR), and other laboratory findings is presented in Fig 2.

Multivariate logistic regression analysis based on measurable admission data (Model I, Table 2) identified several significant predictors of aggravation: age, time of admission, BMI, body temperature, eosinophil, Hb, Ht, AST, γGTP, LDH, and BUN.

Model II, which was constructed using patient interview data (Table 3), identified age, time of admission, impaired consciousness at admission (based on JCS classification), fever at symptom onset, and respiratory distress at onset as significant predictors of aggravation. Notably, common comorbidities such as hypertension, diabetes, renal failure, and active malignancy were not statistically significant in Model II.

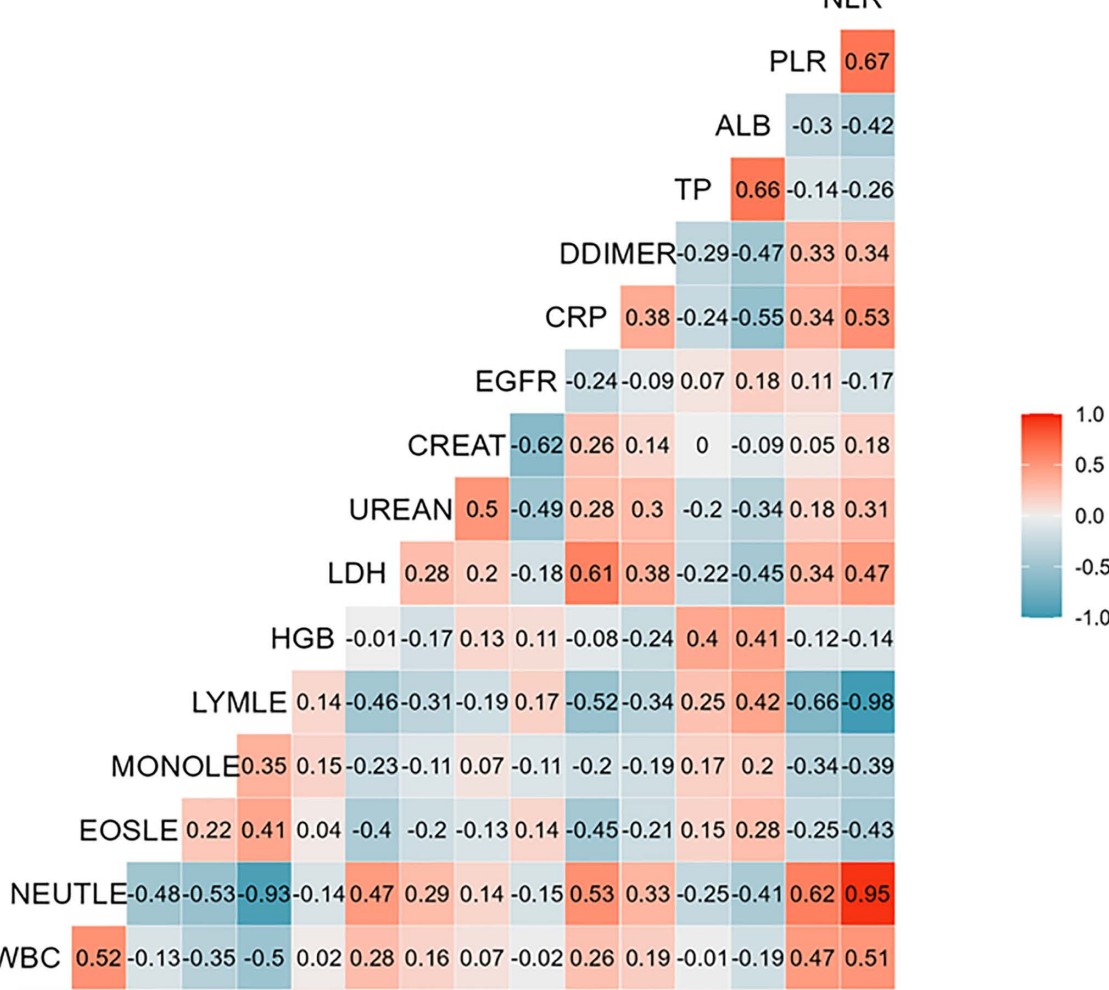

**Fig 2. Correlogram illustrating the correlations between laboratory findings and measurable clinical parameters.** WBC; white blood cells, HGB; hemoglobin, LDH; lactate dehydrogenase, UREAN; blood urea nitrogen, CREAN; creatinine, EGFR; estimated glomerular filtration rate, CRP; C-reactive protein, DDIMER; D-dimer, TP; total protein, ALB; albumin, PLR; platelet-to-lymphocyte ratio, NLR; neutrophil-to-lymphocyte ratio, NEUTLE; neutrophils, EOSLE; eosinophils, MONOLE; monocytes, LYMLE; lymphocytes.

Receiver operating characteristic (ROC) curves for both models are presented in Fig 3 (Panels A and B). The AUC was 0.8928 for Model I and 0.8862 for Model II, indicating good predictive performance for both models.

## Discussion

This study identified risk factors associated with clinical deterioration in hospitalized Japanese patients with COVID-19 during the Wuhan strain pandemic, despite receiving appropriate medical care. The aggravating factors included early admission during the pandemic, older age, impaired consciousness, respiratory distress, elevated body temperature, high BMI, anemia, dehydration, and elevated levels of γ-GTP and LDH. These findings offer a novel contribution to the field by allowing clinicians to recognize patients at higher risk of worsening clinical status, even under standard hospital care.

In contrast to earlier reports, underlying conditions such as diabetes, chronic heart failure, and chronic kidney disease—previously identified as aggravating factors in other populations—did not reach statistical significance in this cohort [4,19,20].

**Table 2. Multivariable Analysis of Aggravating Factors Related to Objective Indicators Based on Measurable Data (Model I, mFAS).**

| Variable | | N | Adjusted odds ratio | 95% confidence interval | P-value |
|---|---|---|---|---|---|
| **Time of admission** | Period I (ref.) | 483 | | | |
| | **Period II** | **716** | **0.372*** | **0.208–0.668** | **0.001** |
| | **Period III** | **1,399** | **0.440*** | **0.276–0.701** | **0.001** |
| **Age (years)** | <65 (ref.) | 1,525 | | | |
| | **65–<75** | **429** | **3.597*** | **2.135–6.058** | **<0.001** |
| | **≥75** | **644** | **3.842*** | **2.219–6.651** | **<0.001** |
| Sex | Male (ref.) | 1,520 | | | |
| | Female | 1,078 | 0.787 | 0.513–1.206 | 0.271 |
| **BMI (kg/m²)** | <25 (ref.) | 1,585 | | | |
| | **≥25** | **770** | **1.610*** | **1.058–2.448** | **0.027** |
| **Body temperature** | | **2,595** | **1.022*** | **1.014–1.030** | **<0.001** |
| SpO₂ | | 2,590 | 0.994 | 0.986–1.002 | 0.099 |
| GCS score | | 2,141 | 0.994 | 0.982–1.006 | 0.351 |
| WBC | | 2,548 | 0.999 | 0.985–1.013 | 0.915 |
| Neutrophils | | 2,474 | 1.049 | 0.993–1.108 | 0.090 |
| **Eosinophils** | | **2,441** | **0.960*** | **0.932–0.988** | **0.007** |
| Monocytes | | 2,133 | 1.009 | 0.991–1.027 | 0.328 |
| Lymphocytes | | 2,481 | 1.024 | 0.975–1.076 | 0.335 |
| RBC | | 2,224 | 0.997 | 0.976–1.019 | 0.772 |
| **Hb** | | **2,546** | **0.960*** | **0.925–0.996** | **0.032** |
| **Ht** | | **2,228** | **1.039*** | **1.001–1.078** | **0.049** |
| MCV | | 2,146 | 1.003 | 0.991–1.015 | 0.613 |
| Plt | | 2,548 | 0.999 | 0.985–1.013 | 0.942 |
| Total protein | | 2,049 | 0.991 | 0.974–1.009 | 0.311 |
| Albumin | | 2,469 | 0.988 | 0.971–1.006 | 0.224 |
| **AST** | | **2,548** | **0.966*** | **0.943–0.989** | **0.003** |
| ALT | | 2,543 | 1.013 | 0.993–1.033 | 0.198 |
| ALP | | 2,177 | 0.987 | 0.968–1.007 | 0.236 |
| **γGTP** | | **2,391** | **1.012*** | **1.002–1.022** | **0.023** |
| **LDH** | | **2,530** | **1.03*** | **1.012–1.049** | **0.001** |
| **BUN** | | **2,540** | **1.015*** | **1.001–1.029** | **0.025** |
| Creatinine | | 2,541 | 1.006 | 0.992–1.02 | 0.411 |
| eGFR | | 1,915 | 1.002 | 0.981–1.024 | 0.886 |
| CRP | | 2,426 | 1.006 | 0.992–1.02 | 0.365 |
| PT-INR | | 2,026 | 1.000 | 0.981–1.02 | 0.993 |
| APTT | | 1,902 | 1.002 | 0.988–1.016 | 0.764 |
| FDP | | 636 | 1.007 | 0.98–1.035 | 0.632 |
| D-dimer | | 2,178 | 0.997 | 0.987–1.007 | 0.498 |

*Statistically significant (p<0.05), BMI; body mass index, GCS; Glasgow Coma Scale, WBC; White blood cells, RBC; Red blood cells, Hb; Hemoglobin, Ht; Hematocrit, MCV; Mean corpuscular volume, Plt; Platelets, AST; Aspartate aminotransferase, ALT; Alanine aminotransferase, ALP; Alkaline phosphatase, γGTP; γ-glutamyl transpeptidase, LDH; Lactate dehydrogenase, BUN; Blood urea nitrogen, eGFR; Estimated glomerular filtration rate, CRP; C-reactive protein, PT-INR; Prothrombin time international normalized ratio, APTT; Activated partial thromboplastin time, FDP; Fibrinogen degradation product

**Table 3. Multivariable Analysis of Aggravating Factors Based on Patient Interviews at Admission and Information from Referring Doctors in Situations without Laboratory Testing (Model II, mFAS).**

| Variable | | N | Adjusted odds ratio | 95% confidence interval | P-value |
|---|---|---|---|---|---|
| **Time of admission** | Period I (ref.) | 483 | | | |
| | **Period II** | **716** | **0.264*** | **0.143–0.486** | **<0.001** |
| | **Period III** | **1,399** | **0.247*** | **0.150–0.407** | **<0.001** |
| **Age (years)** | <65 (ref.) | 1,525 | | | |
| | **65–<75** | **429** | **3.271*** | **1.897–5.640** | **<0.001** |
| | **≥75** | **644** | **3.747*** | **2.131–6.590** | **<0.001** |
| Sex | Female/Male (ref.) | 1,078/1,520 | 0.887 | 0.548–1.436 | 0.626 |
| BMI (kg/m²) | ≥25/<25 (ref.) | 770/1,585 | 1.039 | 0.654–1.650 | 0.871 |
| Smoking | Yes/No (ref.) | 906/1,184 | 1.223 | 0.736–2.031 | 0.442 |
| **JCS category**** | 0 (ref.) | 2,070 | | | |
| | **1,2,3** | **128** | **3.193*** | **1.646–6.193** | **0.001** |
| | **10,20,30** | **26** | **8.077*** | **2.663–24.492** | **<0.001** |
| | **100,200,300** | **8** | **11.681*** | **2.152–63.401** | **0.004** |
| Symptoms at disease onset | | | | | |
| **Fever** | **Yes/No (ref.)** | **1,832/360** | **2.425*** | **1.321–4.453** | **0.004** |
| Olfactory disturbance | Yes/No (ref.) | 138/1,983 | 0.240 | 0.052–1.111 | 0.068 |
| Taste disturbance | Yes/No (ref.) | 175/1,943 | 0.963 | 0.284–3.264 | 0.952 |
| **Respiratory disturbance** | **Yes/No (ref.)** | **258/1,861** | **2.460*** | **1.549–3.906** | **<0.001** |
| Consciousness disturbance | Yes/No (ref.) | 20/2,089 | 0.222 | 0.044–1.120 | 0.072 |
| Underlying disease | | | | | |
| Diabetes | Yes/No (ref.) | 533/1,939 | 1.718 | 0.948–3.111 | 0.076 |
| Hypertension | Yes/No (ref.) | 886/1,702 | 1.791 | 0.972–3.302 | 0.062 |
| Hyperlipidemia | Yes/No (ref.) | 486/1,847 | 1.714 | 0.908–3.235 | 0.096 |
| Renal failure | Yes/No (ref.) | 151/2,129 | 0.578 | 0.225–1.487 | 0.257 |
| Hemodialysis | Yes/No (ref.) | 85/2,192 | 1.554 | 0.517–4.676 | 0.434 |
| Allergic diseases | Yes/No (ref.) | 163/2,100 | 0.633 | 0.159–2.526 | 0.518 |
| Collagen diseases | Yes/No (ref.) | 64/2,226 | 4.336 | 0.111–169.058 | 0.434 |
| Heart valve diseases | Yes/No (ref.) | 41/2,197 | 1.517 | 0.486–4.739 | 0.473 |
| Congestive heart diseases | Yes/No (ref.) | 88/2,161 | 1.966 | 0.959–4.028 | 0.065 |
| Chronic obstructive pulmonary disease | Yes/No (ref.) | 91/2,480 | 4.978 | 0.672–36.897 | 0.122 |
| Interstitial pneumonia | Yes/No (ref.) | 31/2,241 | 1.962 | 0.959–4.012 | 0.065 |
| Pulmonary aspergillosis | Yes/No (ref.) | 6/2,263 | 1.061 | 0.421–2.670 | 0.901 |
| Malignancy, cured | Yes/No (ref.) | 35/2,551 | 0.530 | 0.116–2.433 | 0.418 |
| Gastric cancer, cured | Yes/No (ref.) | 2/2,584 | 4.263 | 0.361–50.282 | 0.250 |
| Pharyngeal and laryngeal cancer, cured | Yes/No (ref.) | 3/2,588 | 2.406 | 0.287–20.139 | 0.422 |
| Malignancy, active | Yes/No (ref.) | 111/2,478 | 5.436 | 0.708–41.737 | 0.104 |
| **Colon cancer, active** | **Yes/No (ref.)** | **8/2,583** | **22.488*** | **1.759–287.436** | **0.017** |
| Prostate cancer, active | Yes/No (ref.) | 12/2,579 | 1.904 | 0.099–36.587 | 0.670 |
| Bladder cancer, active | Yes/No (ref.) | 6/2,585 | 1.659 | 0.085–32.501 | 0.739 |
| **Hepatic cancer, active** | **Yes/No (ref.)** | **8/2,583** | **2.983*** | **1.532–5.809** | **0.001** |
| Multiple myeloma, active | Yes/No (ref.) | 6/2,585 | 2.337 | 0.545–10.027 | 0.254 |
| Treatment for underlying diseases | | | | | |
| Hypertension drugs | Yes/No (ref.) | 807/1,764 | 0.856 | 0.468–1.565 | 0.613 |
| Hyperlipidemia drugs | Yes/No (ref.) | 435/2,145 | 0.556 | 0.283–1.091 | 0.088 |
| Chemotherapy | Yes/No (ref.) | 41/2,538 | 2.921 | 0.693–12.313 | 0.147 |

*(Continued)*

**Table 3.** (Continued)

| Variable | | N | Adjusted odds ratio | 95% confidence interval | P-value |
|---|---|---|---|---|---|
| Diabetes drugs | Yes/No (ref.) | 335/2,243 | 0.924 | 0.477–1.789 | 0.816 |
| Antiplatelet drugs | Yes/No (ref.) | 221/2,372 | 1.275 | 0.759–2.143 | 0.358 |
| **Anticoagulation during admission** | **Yes/No (ref.)** | **545/2,048** | **4.948*** | **3.343–7.323** | **<0.001** |

BMI; body mass index, *Statistically significant (p < 0.05), ** The Japan Coma Scale (JCS) score has been widely used to assess patients' consciousness levels in Japan. JCS scores are divided into four main categories: alert (0), one-, two-, and three-digit codes based on an eye response test, each of which has three subcategories.

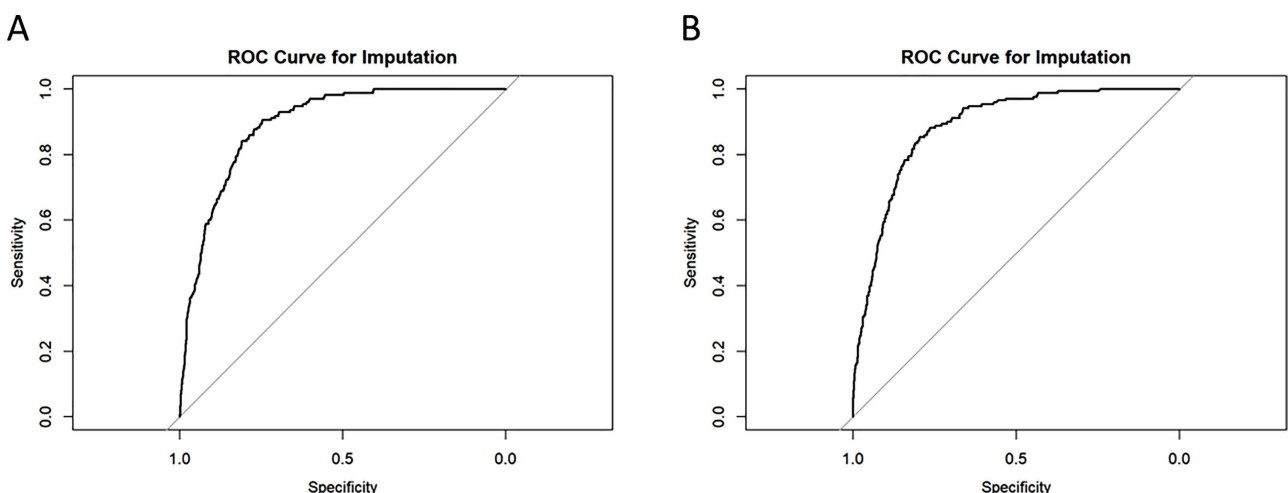

**Fig 3. The area under the receiver operating characteristic (ROC) curves (AUC) for multivariate analysis.** A. Model I. B. Model II.

Risk factors previously associated with high in-hospital mortality during the initial phases of the pandemic [21–23] were also observed in the univariate analysis of this study. However, in the multivariate analysis, the following factors remained statistically significant among those who experienced clinical worsening despite treatment: in Model I (based on measurable data), significant predictors included older age, high BMI, elevated body temperature at admission, anemia, dehydration, and increased γ-GTP and LDH. In Model II (based on patient-reported symptoms), older age, decreased level of consciousness at admission, fever symptom onset, and respiratory symptoms at onset were significant.

A notable finding across both models was the impact of admission timing. Outcomes improved in Periods II and III of the pandemic, indicating that patients admitted during the initial wave experienced higher rates of deterioration. This trend may reflect advancements in therapeutic strategies—such as the introduction of remdesivir or steroids—and the increasing clinical familiarity of healthcare providers with COVID-19 management [19]. Moreover, the reduced time from symptom onset to admission observed in later waves suggests improved awareness among both patients and clinicians.

Established risk factors, including advanced age, male sex, and high BMI, were confirmed in this study [3,4,6,19,21,22,24]. Specifically, patients aged 65–<75 and ≥75 years demonstrated elevated odds of aggravation: 3.60 and 3.84 in Model I, and 3.271 and 3.747 in Model II, respectively. Respiratory impairment and may reflect underlying physiological vulnerability—such as impaired oxygenation capacity or systemic exhaustion—contributing to clinical decline.

Procalcitonin and D-dimer levels were not significantly associated with aggravation in the univariate analysis. CRP, although significant in univariate testing, was not retained as a predictor in the multivariate model. This suggests that CRP, procalcitonin, and D-dimer—while elevated in severe cases [6,19,25,26]—may not predict worsening at admission, as their elevations often reflect disease severity that has already progressed. In contrast, both AST and LDH were significant in the multivariate analysis, with odds ratios of 0.966 and 1.03, respectively. Clinically, elevated LDH, rather than a decline in AST, appears more meaningful for predicting severity. Previous studies have similarly identified LDH as a risk factor for disease progression after admission [6,26].

As of March 10, 2023, Japan's COVID-19 mortality rate stood at 0.2% (72,997 deaths out of 33,320,438 cases), among the lowest worldwide. However, early in the pandemic, mortality rates among hospitalized patients were substantially higher: 21–24% in the United States [27,28], 26% in the United Kingdom [21], and 28% in China [6]. In the present study, the in-hospital mortality rate was 3.8% during the initial Wuhan strain wave and decreased further as the Omicron variant became dominant. While Japan's healthcare system may have contributed to its relatively low overall mortality, it was still essential to identify risk factors for clinical deterioration after admission in order to inform future responses to emerging infectious diseases.

This study has several limitations. First, the predominant viral variant during the study period was the Wuhan strain, with limited representation of the Alpha variant. Therefore, findings may not be generalizable to subsequent variants such as Delta or Omicron. Second, COVID-19 vaccination was not available during the study period, and the overall rate of prior infection was low, meaning immune responses to SARS-CoV-2 likely differed from later stages of the pandemic. Third, clinical data were collected from the time of hospital admission onward; thus, variability in the time between symptom onset and admission may have influenced results. The study did not capture the complete disease trajectory from onset to recovery but instead focused on in-hospital deterioration. Although cases with severe disease on admission were excluded, this methodological choice may have limited the ability to fully identify predictive factors for worsening. However, given that all diagnosed COVID-19 cases in Japan were required by regulation to be hospitalized during the study period, it is assumed that very few patients were managed solely as outpatients from onset to recovery.

## Conclusion

This study investigated the factors associated with clinical deterioration in Japanese patients hospitalized with COVID-19 during the Wuhan strain phase of the pandemic, despite receiving appropriate medical care. The identified aggravating factors included early admission during the initial pandemic wave, older age, elevated BMI, high body temperature, anemia, dehydration and elevated levels of γ-GTP and LDH. These findings offer valuable insight into patient groups at higher risk of severe illness despite treatment, particularly in the context of future emerging respiratory viral infections. This information may contribute to pandemic preparedness and clinical risk stratification strategies. Further research is needed to assess whether these risk factors also apply to other emerging infectious diseases, such as novel strains of influenza.

## Supporting information

**S1 Table. Univariate analysis of aggravating factors related to patient demographics (mFAS).** (DOCX)

**S2 Table. Univariate analysis of symptom onset and physical findings on admission (mFAS).** (DOCX)

**S3 Table. Univariate analysis of laboratory findings on admission (mFAS).** (DOCX)

**S4 Table. List of 36 institutions across Japan that joined the JAID-COVID registry.** (DOCX)

## Acknowledgments

We extend our sincere gratitude to the facility research managers of the 36 participating hospitals throughout Japan in the COVID registry of the JAID, listed in S4 Table. The registry was managed by the Ad Hoc Committee for the COVID-19–Aggravating Factors Search Project of JAID, chaired by Kazuyoshi Kawakami (former professor, Tohoku University; currently at Hirose Hospital; kazuyoshi.kawakami.a1@tohoku.ac.jp). Additional committee members include Itaru Nakamura (Tokyo Medical University Hospital), Yusuke Koizumi (Wakayama Medical University), Hideki Araoka (Toranomon Hospital), and Hiroaki Hata (National Hospital Organization Kyoto Medical Center). Some participating hospitals submitted data from a nationwide COVID-19 inpatient registry, COVID-19 Registry Japan (COVIREGI-JP), which operates under the Repository of Data and Biospecimen of Infectious Disease (REBIND) project commissioned by the Ministry of Health, Labour and Welfare of Japan. We thank Translational Research Center for Medical Innovation for conducting the statistical analysis.

## Author contributions

**Conceptualization:** Itaru Nakamura.

**Data curation:** Itaru Nakamura, Yusuke Koizumi, Hideki Araoka, Hiroaki Hata, Kazuyoshi Kawakami.

**Methodology:** Takashi Miki, Yusuke Tanaka, Chie Kobayashi.

**Supervision:** Kazuyoshi Kawakami.

**Validation:** Takashi Miki, Yusuke Tanaka, Chie Kobayashi.

**Writing – original draft:** Itaru Nakamura.

**Writing – review & editing:** Yusuke Koizumi, Hideki Araoka, Hiroaki Hata, Kazuyoshi Kawakami.

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
