## [Editor Report · Decision Letter 0]

13 Mar 2024

Dear Dr. Araoka,

Thank you for submitting your manuscript to PLOS ONE. After careful consideration, we feel that it has merit but does not fully meet PLOS ONE’s publication criteria as it currently stands. Therefore, we invite you to submit a revised version of the manuscript that addresses the points raised during the review process.

There are many similar publications related with this topic. 

I suggest some points for improvement.

- Please highlight the prognostic factors in your conclusion abstract. This study determined the aggravating factors for severe COVID-19 development despite

medical care after admission in Japanese people during the Wuhan clone pandemic.Please be detailed in the aggravating factors. 

- The method should be detail. Is this retrospective cohort ?

- Please highlight the table.

- Please begin the discussion with the main findings of your study. 

- Do the authors perform adjusted analysis? Please elaborate more.

- The conclusion should be concise and clear. 

We look forward to receiving your revised manuscript.

Kind regards,

Rizaldy Taslim Pinzon

Academic Editor

PLOS ONE

Journal Requirements:

   "This JAID-COVID registry study was conducted as a JAID joint research project with research funding from Astellas Pharma."

6. One of the noted authors is a group or consortium "the COVID registry of the Japanese Association for Infectious Diseases". In addition to naming the author group, please list the individual authors and affiliations within this group in the acknowledgments section of your manuscript. Please also indicate clearly a lead author for this group along with a contact email address.

7. Please include your tables as part of your main manuscript and remove the individual files. Please note that supplementary tables (should remain/ be uploaded) as separate "supporting information" files

Additional Editor Comments:

There are many similar publications related with this topic.

I suggest some points for improvement.

- Please highlight the prognostic factors in your conclusion abstract. This study determined the aggravating factors for severe COVID-19 development despite

medical care after admission in Japanese people during the Wuhan clone pandemic.Please be detailed in the aggravating factors.

- The method should be detail. Is this retrospective cohort ?

- Please highlight the table.

- Please begin the discussion with the main findings of your study.

- Do the authors perform adjusted analysis? Please elaborate more.

- The conclusion should be concise and clear.

---

## [Author Response · Author response to Decision Letter 1]

28 Mar 2024

There are many similar publications related with this topic.

I suggest some points for improvement.

- Please highlight the prognostic factors in your conclusion abstract. This study determined the aggravating factors for severe COVID-19 development despite medical care after admission in Japanese people during the Wuhan clone pandemic. Please be detailed in the aggravating factors.

Ans.) Thank you for this suggestion. We have modified the abstract to clarify the aggravating factors.

“Results

A total of 2,884 patients were enrolled at 36 institutions throughout Japan. After excluding 25 ineligible patients and 45 protocol deviations, 2,814 patients were analyzed. In multivariate analysis to explore aggravating factors, the time of admission, age, body mass index (BMI), body temperature, hemoglobin, hematocrit, eosinophil, γGTP, LDH, and BUN were identified as factors.

Conclusion

This study determined the aggravating factors for severe COVID-19 development despite medical care after admission in Japanese people during the Wuhan clone pandemic. The aggravating factors were admission at the beginning of the pandemic, older age, high BMI, high body temperature, anemia, cholestasis, and dehydration.”

- The method should be detail. Is this retrospective cohort?

Ans.) Thank you for the suggestion. This was a retrospective cohort study. We have modified the abstract and the main text to clarify the study design.

Abstract

“Materials and methods

This was a retrospective, multicenter cohort study using the Japanese Association for Infectious Diseases COVID registry to explore aggravating factors for severe COVID-19 development despite medical care after admission in Japanese people. The study population comprised Japanese patients definitively diagnosed with COVID-19 between January 2020 and March 2021, the period of the Wuhan clone pandemic.”

Main body

“Patients and study design

This was a retrospective, multicenter cohort study of aggravating factors in Japanese COVID-19 patients. It used the COVID registry of the Japanese Association for Infectious Diseases (JAID), joined by 36 institutions throughout Japan.”

- Please highlight the table.

Ans.) Thank you for the suggestion. We have highlighted all columns with statistical significance using bold font.

- Please begin the discussion with the main findings of your study.

Ans.) Thank you for this suggestion. We have modified the abstract to clarify the aggravating factors.

“Discussion

This study determined the aggravating factors for severe COVID-19 development despite medical care after admission in Japanese people during the Wuhan clone pandemic. The aggravating factors were identified as admission at the beginning of the pandemic, older age, high BMI, high body temperature, anemia, cholestasis, and dehydration.”

- Do the authors perform adjusted analysis? Please elaborate more.

Ans.) Thank you for your comments regarding the adjusted analysis. We apologize for not making this clear in the manuscript. We performed adjusted analyses using two multivariate logistic regression models to examine the association between various variables and the risk of aggravation.

The first model (Model I) included objective indicators such as measurable data from laboratory findings or objective data that could be obtained even if a patient interview was not possible at admission. The second model (Model II) included information obtained from patient interviews at admission and from referring doctors when laboratory testing was not available.

For both models, we selected variables with a p-value of <0.1 and a missing rate of <30% based on the univariate analysis, as well as clinically important variables known to be associated with the outcome. We employed multiple imputation to handle missing data.

The adjusted odds ratios (ORs) and their 95% confidence intervals (CIs) were reported for each variable included in the models. We have modified the manuscript to provide a clearer explanation of the adjusted analysis in the Statistical analysis section.

Statistical analysis

“In addition, two multivariate logistic regression models for FAS 2 were used to examine the regression coefficient (OR) of each variable for aggravation. The first model (Model I) estimated the risk of aggravation based on objective indicators such as measurable data from laboratory findings or objective data, even if a patient interview could not be performed at admission. The second model (Model II) estimated the risk of aggravation based on patient interviews at admission and information from referring doctors in situations without laboratory testing. For both models, variables with a p-value of <0.1 and a missing rate of <30% based on the univariate analysis and known clinically important variables were included. The adjusted OR and 95% CI were reported. Multiple imputation was adopted for missing data. P-values of less than 0.05 were considered statistically significant.”

- The conclusion should be concise and clear.

Ans.) Thank you for this suggestion. We have modified the conclusions of the abstract and main text to clarify the aggravating factors.

Abstract

“Conclusion

This study determined the aggravating factors for severe COVID-19 development despite medical care after admission in Japanese people during the Wuhan clone pandemic. The aggravating factors were admission at the beginning of the pandemic, older age, high BMI, high body temperature, anemia, cholestasis, and dehydration.”

Body

“Conclusion

This study examined the aggravating factors for severe COVID-19 development despite medical care after admission in Japanese people during the Wuhan clone pandemic period. The aggravating factors were identified as admission at the beginning of the pandemic, older age, high BMI, high body temperature, anemia, cholestasis, and dehydration.”

Journal Requirements:

　　　Ans). We apologize for not following these journal requirements. We have made all the required modifications.

Ans.) Thank you for your comments. Because the need for consent was waived by the ethics committee, we have added the following content.

Main text

“Ethical considerations

This study was approved by the Tohoku University Ethics Committee and the institutional review boards of the collaborating institutes (UMIN registration number 000043802). The need for participant consent was formally waived by the Ethics Committee because the cohort study was retrospective and exploratory.”

“Ethics approval and consent to participate

This study was approved by the Ethics Committee of Tohoku University (approval number T2020-0161). The need for participant consent was formally waived by the Ethics Committee. This study was performed following the Declaration of Helsinki of 1964 and its later amendments.”

Ans). We apologize for not meeting this journal requirement. We have deleted the funding information from the main text.

"This JAID-COVID registry study was conducted as a JAID joint research project with research funding from Astellas Pharma. "Please state what role the funders took in the study. If the funders had no role, please state: "The funders had no role in study design, data collection and analysis, decision to publish, or preparation of the manuscript." If this statement is not correct you must amend it as needed. Please include this amended Role of Funder statement in your cover letter; we will change the online submission form on your behalf.

　　Ans.) Thank you for these comments. The funders had a role in the methodology and data validation. We have included this information in the cover letter.

　　Ans.) I have updated my ORCID iD information.

6. One of the noted authors is a group or consortium "the COVID registry of the Japanese Association for Infectious Diseases". In addition to naming the author group, please list the individual authors and affiliations within this group in the acknowledgments section of your manuscript. Please also indicate clearly a lead author for this group along with a contact email address.

Ans.) I have now added information on the Ad Hoc Committee managing the COVID registry of the Japanese Association for Infectious Diseases.

“Acknowledgments

Regarding data registration, we express our gratitude to the facility research managers of the 36 participating hospitals throughout Japan in the COVID registry of the JAID, listed in Table S4. The Ad Hoc Committee for the COVID-19–Aggravating Factors Search Project of the JAID managed this registry, led by Kazuyoshi Kawakami (Former professor of Tohoku University, Hirose Hospital, kawakami@med.tohoku.ac.jp). The other members are Itaru Nakamura (Tokyo Medical University Hospital), Yusuke Koizumi (Wakayama Medical University), Hideki Araoka (Toranomon Hospital), and Hiroaki Hata (National Hospital Organization Kyoto Medical Center).”

7. Please include your tables as part of your main manuscript and remove the individual files. Please note that supplementary tables (should remain/ be uploaded) as separate "supporting information" files

Ans.) We apologize for not following the journal requirement. We have now added the tables in the main text.

Ans.) We apologize for not meeting this journal requirement. We have deleted the supplemental table and added the supporting information in the main text.

---

## [Decision Letter · Decision Letter 1]

31 Mar 2025

Dear Dr. Araoka,

Thank you for submitting your manuscript to PLOS ONE. After careful consideration, we feel that it has merit but does not fully meet PLOS ONE’s publication criteria as it currently stands. Therefore, we invite you to submit a revised version of the manuscript that addresses the points raised during the review process.

The reviewers suggest significant revisions to your manuscript.  Therefore, I invite you to respond to the reviewers' comments and revise your manuscript.

We look forward to receiving your revised manuscript.

Kind regards,

Fumihiro Yamaguchi

Academic Editor

PLOS ONE

Reviewers' comments:

Reviewer's Responses to Questions

**Comments to the Author**

Reviewer #1: (No Response)

Reviewer #2: (No Response)

2. Is the manuscript technically sound, and do the data support the conclusions?

Reviewer #1: Partly

Reviewer #2: No

3. Has the statistical analysis been performed appropriately and rigorously?

Reviewer #1: No

Reviewer #2: No

4. Have the authors made all data underlying the findings in their manuscript fully available?

Reviewer #1: Yes

Reviewer #2: Yes

5. Is the manuscript presented in an intelligible fashion and written in standard English?

Reviewer #1: Yes

Reviewer #2: Yes

Reviewer #1: In this manuscript, the authors aim to evaluate the aggravating factors contributing to severe COVID-19 progression despite medical care after hospital admission in Japanese individuals during the Wuhan strain pandemic, considering various risk factors. The study addresses an important issue regarding the use of laboratory parameters to assess disease worsening after admission. However, the manuscript presents a rather superficial analysis rather than serving as a robust reference. In particular, the study has critical flaws in its design, particularly in the methodology, and lacks important data. The authors should address the following points:

1. While the introduction provides an acceptable scientific background, it would be beneficial to better highlight the distinctiveness of the study and discuss other potential confounders not included in the analysis.

2. Another major concern is why the authors did not include age, sex, BMI, smoking status, and other risk factors alongside laboratory findings in the composite models, especially given the results of the univariate analysis. Additionally, why did the authors not use binary logistic regression with forward stepwise selection to assess independent predictors?

3. How did the authors evaluate the correlation between PLR or NLR and the laboratory findings, as well as other biomarkers such as WBC, HGB, ALB, and D-dimer? I suggest performing a Spearman’s correlation analysis including all factors and presenting a correlation matrix, such as a correlogram or heatmap, in the Results section.

4. I recommend assessing the discriminatory power of laboratory variables in predicting disease progression and mortality in COVID-19 patients by calculating the area under the receiver operating characteristic (ROC) curves (AUC). Using composite models could significantly improve the predictive accuracy (AUC values) for disease aggravation, thereby adding value to the manuscript.

5. Although the authors discuss the inflammatory response, it would be helpful to further elaborate on why CRP, D-dimer, and procalcitonin were not statistically significant factors in this study.

6. The Conclusion section requires improvement to better highlight the major findings of the study in relation to similar research. Additionally, future research directions should be briefly but clearly outlined.

By addressing these concerns, the manuscript could be significantly strengthened in terms of scientific rigor and impact.

Reviewer #2: The manuscript examined the aggravating factors for severe COVID-19 development despite medical care after admission among Japanese people during the COVID-19 pandemic. The topic is interesting, but the following points should be considered by the authors.

1. The three sets or groups of analysis population FAS, FAS1, and FAS2 are really confusing; the authors need to make this point clear.

2. Line 180-182 (A total of 2,884 patients were enrolled at 36 institutions throughout Japan that joined the JAID181 COVID registry. After excluding 25 ineligible patients and 45 cases of protocol deviation, 2,814 patients were analyzed), then the authors jumped to discussing about FAS, FAS1, and FAS2.

I do not know which one is the study sample?

If you have three study samples, then you could compare the parameters in your study samples.

3. The authors must compare aggravating factors such as age, sex, body temperature, and laboratory data between patients who were discharged from the hospital and those who died from the disease. In this case, you will be able to prove which factor significantly contributed as an aggravating factor.

4. The authors have to explain which medical care the patients had received.

I think the authors have to re-perform the statistical analysis appropriately and rigorously so that the tables are presented in an intelligible fashion and make sense.

**Do you want your identity to be public for this peer review?** For information about this choice, including consent withdrawal, please see our Privacy Policy

Reviewer #1: No

Reviewer #2: No

---

## [Author Response · Author response to Decision Letter 2]

18 Jun 2025

Response to reviewers

Reviewer #1:

In this manuscript, the authors aim to evaluate the aggravating factors contributing to severe COVID-19 progression despite medical care after hospital admission in Japanese individuals during the Wuhan strain pandemic, considering various risk factors. The study addresses an important issue regarding the use of laboratory parameters to assess disease worsening after admission. However, the manuscript presents a rather superficial analysis rather than serving as a robust reference. In particular, the study has critical flaws in its design, particularly in the methodology, and lacks important data. The authors should address the following points:

→ We appreciate your detailed and constructive feedback. We have carefully considered each point and revised the manuscript accordingly. Below, we provide specific responses and indicate where corresponding revisions have been made in the manuscript.

1. While the introduction provides an acceptable scientific background, it would be beneficial to better highlight the distinctiveness of the study and discuss other potential confounders not included in the analysis.

→Thank you for this valuable suggestion. Previous studies have primarily identified risk factors in patients who were already in severe condition at the time of hospitalization or when they initially received medical care. In contrast, our study is distinctive in its focus on identifying risk factors for clinical deterioration despite the prevision of appropriate medical treatment. To better reflect this unique contribution, we have revised and streamlined the introduction, removed redundant background information on COVID-19, and emphasized our study’s novel focus.

Revised text (Lines 53–75):

Introduction

“Coronavirus disease 2019 (COVID-19) is caused by severe acute respiratory syndrome coronavirus 2 (SARS-CoV-2). The global mortality rate was particularly high at the onset of the pandemic, dominated by the Wuhan strain and subsequently by the Alpha and Delta variants. Mortality rates in the Japanese population have been reported to be among the lowest worldwide. However, during the Wuhan strain period—when vaccines were not yet available—some patient groups in Japan experienced disease worsening associated with common factors and comorbidities.

Previous studies from multiple countries have identified various factors influencing COVID-19 prognosis. Reported risk factors include age, sex, number and type of comorbidities, respiratory rate, body mass index (BMI), peripheral oxygen saturation, and level of consciousness, as measured by tools such as the Glasgow Coma Scale (GCS). Laboratory parameters reported as prognostic indicators include white blood cell (WBC) count, lymphocyte count, blood glucose, albumin (ALB), aspartate aminotransferase (AST), lactate dehydrogenase (LDH), creatinine kinase, urea, C-reactive protein (CRP), troponin I, procalcitonin, and D-dimer (1–10). However, these risk factors have generally been associated with patients already in severe condition upon hospital admission rather than with disease aggravation occurring despite medical care after admission. Moreover, most studies investigating risk factors for severe disease progression or death in COVID-19 patients have focused on non-Japanese populations. Data on COVID-19 outcomes specific to Japanese patients remain limited to a few reports (11–14).

In this study, we aimed to identify factors associated with COVID-19 aggravation during inpatient treatment despite receiving medical care, focusing on the Wuhan strain–dominant period in an unvaccinated Japanese population.”

2. Another major concern is why the authors did not include age, sex, BMI, smoking status, and other risk factors alongside laboratory findings in the composite models, especially given the results of the univariate analysis. Additionally, why did the authors not use binary logistic regression with forward stepwise selection to assess independent predictors?

→ Thank you for your important comment. We would like to clarify that age, sex, BMI, smoking status, and several laboratory variables were included in both the univariate and multivariate analyses. These results were provided in Supplementary Tables 1 and 3. However, we acknowledge that the supplemental placement may have limited their visibility. Based on the univariate analysis, variables with statistically significant associations—such as WBC, hemoglobin, albumin, LDH, and renal function—were included in the multivariate model. Although BMI did not reach statistical significance in the univariate analysis, we included it in the multivariate model due to its well-established clinical importance.

Regarding your suggestion to use binary logistic regression with forward stepwise selection, we did consider this approach during our initial analysis planning. However, our dataset contained missing values with varying degrees of completeness across variables. We were concerned that an automatic stepwise method might yield unstable or biased results depending on the variable inclusion sequence and missing data distribution. Therefore, we opted for a full model, selecting variables based on clinical relevance and p-values, to maintain model stability and interpretability.

Revised text (Lines 156–160):

“Although the forward stepwise method is commonly used for variable selection, the presence of variable missingness and varying missing rates raised concerns about model instability and bias from automated selection. Therefore, variables considered clinically important or with significant p-values were selected a priori, and a full-model approach including all these variables was adopted.”

3. How did the authors evaluate the correlation between PLR or NLR and the laboratory findings, as well as other biomarkers such as WBC, HGB, ALB, and D-dimer? I suggest performing a Spearman’s correlation analysis including all factors and presenting a correlation matrix, such as a correlogram or heatmap, in the Results section.

→ Thank you for your insightful suggestion. In response, we calculated the platelet-to-lymphocyte ratio (PLR) and neutrophil-to-lymphocyte ratio (NLR) for Model 1 of FAS2 cohort (Table 2 in the manuscript). We then created a correlogram using these and all clinical test-related variables. This explanation has also been added to the revised Methods and Results section.

Revised text (Lines 143–146):

Statistical Analysis

“Spearman’s correlation analysis was performed to evaluate associations between platelet-to-lymphocyte ratio (PLR) (16, 17), neutrophil-to-lymphocyte ratio (NLR) (16, 18), and laboratory findings, with correlograms constructed for all clinical test variables.”

Revised text (Lines 228–243):

Results

“The correlogram showing the correlations between the platelet-to-lymphocyte ratio (PLR), neutrophil-to-lymphocyte ratio (NLR), and other laboratory findings is presented in Figure 2.

Figure 2. Correlogram illustrating the correlations between laboratory findings and measurable clinical parameters.

Figure legend: GCS; Glasgow Coma Scale, WBC; white blood cells, RBC; red blood cells, HGB; hemoglobin, HCT; hematocrit, PLAT; platelets, AST; aspartate aminotransferase, ALT; alanine aminotransferase, ALP; alkaline phosphatase, GGT; γ-glutamyl transpeptidase, LDH; lactate dehydrogenase, UREAN; blood urea nitrogen, CREAN; creatinine, EGFR; estimated glomerular filtration rate, CRP; C-reactive protein, INR; international normalized ratio, APTT; activated partial thromboplastin time, FDP; fibrinogen degradation products, DDIMER; D-dimer, TP; total protein, ALB; albumin, PLR; platelet-to-lymphocyte ratio, NLR; neutrophil-to-lymphocyte ratio, TMP; temperature, NEUTLE; neutrophils, EOSLE; eosinophils, MONOLE; monocytes, LYMLE; lymphocytes.”

4. I recommend assessing the discriminatory power of laboratory variables in predicting disease progression and mortality in COVID-19 patients by calculating the area under the receiver operating characteristic (ROC) curves (AUC). Using composite models could significantly improve the predictive accuracy (AUC values) for disease aggravation, thereby adding value to the manuscript.

→ Thank you for pointing this out. We assessed the discriminatory power of Model I and Model II by generating receiver operating characteristic (ROC) curves and calculating the area under the curve (AUC) for each model. This explanation has been added to the Methods and Results section.

Revised text (Lines 36–38, Lines 45-46):

Abstract

“The predictive performance of two multivariate models—Model I (based on measurable clinical findings) and Model II (based on interview data)—was assessed using the area under the receiver operating characteristic curve (AUC).”

“The AUCs were 0.8928 for Model I and 0.8862 for Model II, indicating strong discriminatory power.”

Revised text (Lines 153–155):

Statistical Analysis

“The area under the receiver operating characteristic (ROC) curve (AUC) was calculated to evaluate the discriminatory power of each model.”

Revised text (Lines 252–254, Lines266-269):

Results

“Receiver operating characteristic (ROC) curves for both models are presented in Figure 3 (Panels A and B). The AUC was 0.8928 for Model I and 0.8862 for Model II, indicating good predictive performance for both models.”

“Figure 3.　The area under the receiver operating characteristic (ROC) curves (AUC) for multivariate analysis.

Figure 3-A. Model I

Figure 3-B. Model II”

5. Although the authors discuss the inflammatory response, it would be helpful to further elaborate on why CRP, D-dimer, and procalcitonin were not statistically significant factors in this study.

→ Thank you for this important comment. In our study, we excluded cases that were already severe at the time of admission as well as those that became severe on the same day. This allowed us to focus on patients who were non-severe upon admission and to identify risk factors associated with clinical deterioration despite standard medical care. The lack of statistical significance for CRP, D-dimer, and procalcitonin may be due to the fact that these markers are more reflective of an existing state, rather than predictive of future clinical worsening.

Revised text (Lines 303-307):

Discussion

“Procalcitonin and D-dimer levels were not significantly associated with aggravation in the univariate analysis. CRP, although significant in univariate testing, was not retained as a predictor in the multivariate model. This suggests that CRP, procalcitonin, and D-dimer—while elevated in severe cases (6, 19, 25, 26)—may not predict worsening at admission, as their elevations often reflect disease severity that has already progressed.”

6. The Conclusion section requires improvement to better highlight the major findings of the study in relation to similar research. Additionally, future research directions should be briefly but clearly outlined. By addressing these concerns, the manuscript could be significantly strengthened in terms of scientific rigor and impact.

→ Thank you for your thoughtful feedback. Based on the results of this study, we have added the following points to consider in the future research and what might be expected during future pandemics caused by emerging infectious diseases.

Revised text (Lines 48-51):

Abstract

“This study identified key risk factors for COVID-19 progression despite inpatient medical care among Japanese patients. These findings underscore the importance of early risk stratification for patients at high risk of deterioration despite treatment and may inform preparedness strategies for future respiratory pandemics.”

Revised text (Lines 336-345):

Conclusion

“This study investigated the factors associated with clinical deterioration in Japanese patients hospitalized with COVID-19 during the Wuhan strain phase of the pandemic, despite receiving appropriate medical care. The identified aggravating factors included early admission during the initial pandemic wave, older age, elevated BMI, high body temperature, anemia, dehydration and elevated levels of γ-GTP and LDH. These findings offer valuable insight into patient groups at higher risk of severe illness despite treatment, particularly in the context of future emerging respiratory viral infections. This information may contribute to pandemic preparedness and clinical risk stratification strategies. Further research is needed to assess whether these risk factors also apply to other emerging infectious diseases, such as novel strains of influenza.”

Reviewer #2: The manuscript examined the aggravating factors for severe COVID-19 development despite medical care after admission among Japanese people during the COVID-19 pandemic. The topic is interesting, but the following points should be considered by the authors.

→Thank you for your valuable comments. We have carefully addressed each of your points below and revised the manuscript accordingly to clarify and strengthen our analysis.

1. The three sets or groups of analysis population FAS, FAS1, and FAS2 are really confusing; the authors need to make this point clear.

→ Thank you for pointing this out. We agree that referring to FAS, FAS1, and FAS2 could be confusing. In the revised manuscript, we have simplified the terminology and focused our analysis on the FAS2 dataset. Specifically, we now refer to the analytical cohort as the modified full analysis set (mFAS) instead of FAS2. This change is reflected in the revised figure and throughout the text. The original dataset was designated the full analysis set (FAS), and after excluding patients who met the exclusion criteria or who were already in severe condition at the time of admission (or became severe on the day of admission), we defined the resulting cohort as mFAS, which was used for all statistical analyses.

Revised text (Lines 134-142):

Statistical Analysis

“The full analysis set (FAS) included all registered patients after applying exclusion criteria. For statistical analysis, a modified full analysis set (mFAS) was created by excluding patients who exhibited disease aggravation on the day of admission or before admission. Background characteristics and risk factors for worsening after admission were analyzed using the mFAS, as the focus was on predicting outcomes among patients who were not initially severe but who worsened despite medical care.

Univariate logistic regression models were applied to the mFAS to estimate odds ratios (ORs) and 95% confidence intervals (CIs) for each explanatory variable regarding the presence or absence of disease aggravation.”

Revised text (Lines 147-150):

“Two multivariate logistic regression models were developed using mFAS to assess the association of variables with aggravation risk. Model I estimated risk based on objective measures such as laboratory data and other measurable variables available at admission without requiring patient interviews.”

2. Line 180-182 (A total of 2,884 patients were enrolled at 36 institutions throughout Japan that joined the JAID181 COVID registry. After excluding 25 ineligible patients and 45 cases of protocol deviation, 2,814 patients were analyzed), then the authors jumped to discussing about FAS, FAS1, and FAS2. I do not know which one is the study sample? If you have three study samples, then you could compare the parameters in your study samples.

→ Thank you for highlighting this point. We acknowledge that the transition between data descriptions may have been unclear. As noted above, we have revised the figure and the Methods section to clearly define the progression from FAS to mFAS. The FAS included all eligible patients (n = 2,814), while the mFAS was derived by excluding patients who were already in a severe condition or worsened on the day of admission. This dataset (mFAS) serves as the primary sample for our analysis.

Revised text (Lines 171-178):

Results

“A total of 2,884 patients were enrolled from 36 institutions across Japan participating in the JAID-COVID registry. After excluding 25 ineligible patients and 45 with protocol deviations, 2,814 cases were included in the full analysis set (FAS). To identify factors associated with clinical aggravation despite medical care, patients who were already severe before adm

---

## [Decision Letter · Decision Letter 2]

12 Oct 2025

Risk factors for aggravated COVID-19 despite medical care after admission among Japanese patients: A Japanese Association for Infectious Diseases COVID registry study

PONE-D-24-06564R2

Dear Dr. Araoka,

We’re pleased to inform you that your manuscript has been judged scientifically suitable for publication and will be formally accepted for publication once it meets all outstanding technical requirements.

Kind regards,

Fumihiro Yamaguchi

Academic Editor

PLOS ONE

Additional Editor Comments (optional):

Reviewers' comments:

Reviewer's Responses to Questions

**Comments to the Author**

Reviewer #3: All comments have been addressed

Reviewer #4: All comments have been addressed

Reviewer #5: All comments have been addressed

2. Is the manuscript technically sound, and do the data support the conclusions?

Reviewer #3: Yes

Reviewer #4: Yes

Reviewer #5: Yes

3. Has the statistical analysis been performed appropriately and rigorously?

Reviewer #3: (No Response)

Reviewer #4: Yes

Reviewer #5: Yes

4. Have the authors made all data underlying the findings in their manuscript fully available?

Reviewer #3: No

Reviewer #4: Yes

Reviewer #5: Yes

5. Is the manuscript presented in an intelligible fashion and written in standard English?

Reviewer #3: Yes

Reviewer #4: Yes

Reviewer #5: Yes

Reviewer #3: (No Response)

Reviewer #4: Thank you for your efforts in revising the manuscript in line with the previous reviewer comments. The language has been improved, and the content has been clarified in a more systematic and coherent manner.

While the topic has been extensively explored in the literature, your presentation is clear and well-structured. I appreciate your work and wish you success with the editorial evaluation.

Best regards.

Reviewer #5: The authors have adequately responded to all reviewer comments, and the revisions have improved the quality of the manuscript.

**Do you want your identity to be public for this peer review?** For information about this choice, including consent withdrawal, please see our Privacy Policy

Reviewer #3: No

Reviewer #4: No

Reviewer #5: **Yes: ** Samrad Mehrabi

---

## [Editor Report · Acceptance letter]

PONE-D-24-06564R2

PLOS ONE

Dear Dr. Araoka,

I'm pleased to inform you that your manuscript has been deemed suitable for publication in PLOS ONE. Congratulations! Your manuscript is now being handed over to our production team.

Kind regards,

on behalf of

Dr. Fumihiro Yamaguchi

Academic Editor

PLOS ONE